# Molecular Epidemiology of HIV-1 Infected Migrants Followed Up in Portugal: Trends between 2001–2017

**DOI:** 10.3390/v12030268

**Published:** 2020-02-28

**Authors:** Victor Pimentel, Marta Pingarilho, Daniela Alves, Isabel Diogo, Sandra Fernandes, Mafalda Miranda, Andrea-Clemencia Pineda-Peña, Pieter Libin, M. Rosário O. Martins, Anne-Mieke Vandamme, Ricardo Camacho, Perpétua Gomes, Ana Abecasis

**Affiliations:** 1Global Health and Tropical Medicine (GHTM), Instituto de Higiene e Medicina Tropical/Universidade Nova de Lisboa (IHMT/UNL), 1349-008 Lisboa, Portugal; victor.pimentel@ihmt.unl.pt (V.P.); martapingarilho@ihmt.unl.pt (M.P.); daniela.alves@ihmt.unl.pt (D.A.); a21000919@ihmt.unl.pt (M.M.); andreapinedap@gmail.com (A.-C.P.-P.); mrfom@ihmt.unl.pt (M.R.O.M.); annemie.vandamme@kuleuven.be (A.-M.V.); 2Laboratório de Biologia Molecular (LMCBM, SPC, CHLO-HEM), 1349-019 Lisboa, Portugal; ifmadeira@chlo.min-saude.pt (I.D.); smfernandes@chlo.min-saude.pt (S.F.); pcrsilva@chlo.min-saude.pt (P.G.); 3Molecular Biology and Immunology Department, Fundación Instituto de Inmunología de Colombia, Basic Sciences Department, School of Medicine and Health Sciences, Universidad del Rosario, Bogotá 111321, Colombia; 4KU Leuven, Clinical and Epidemiological Virology, Department of Microbiology, Immunology and Transplantation, Rega Institute for Medical Research, 3000 Leuven, Belgium; pieter.libin@vub.ac.be (P.L.); ricardojorge.camacho@kuleuven.be (R.C.); 5Artificial Intelligence lab, Department of computer science, Vrije Universiteit Brussel, 1050 Brussels, Belgium; 6Centro de Investigação Interdisciplinar Egas Moniz (CiiEM), Instituto Superior de Ciências da Saúde Egas Moniz, 2829-511 Caparica, Portugal

**Keywords:** migrants, molecular epidemiology, HIV drug resistance mutations

## Abstract

Migration is associated with HIV-1 vulnerability. Objectives: To identify long-term trends in HIV-1 molecular epidemiology and antiretroviral drug resistance (ARV) among migrants followed up in Portugal Methods: 5177 patients were included between 2001 and 2017. Rega, Scuel, Comet, and jPHMM algorithms were used for subtyping. Transmitted drug resistance (TDR) and Acquired drug resistance (ADR) were defined as the presence of surveillance drug resistance mutations (SDRMs) and as mutations of the IAS-USA 2015 algorithm, respectively. Statistical analyses were performed. Results: HIV-1 subtypes infecting migrants were consistent with the ones prevailing in their countries of origin. Over time, overall TDR significantly increased and specifically for Non-nucleoside reverse transcriptase inhibitor (NNRTIs) and Nucleoside reverse transcriptase inhibitor (NRTIs). TDR was higher in patients from Mozambique. Country of origin Mozambique and subtype B were independently associated with TDR. Overall, ADR significantly decreased over time and specifically for NRTIs and Protease Inhibitors (PIs). Age, subtype B, and viral load were independently associated with ADR. Conclusions: HIV-1 molecular epidemiology in migrants suggests high levels of connectivity with their country of origin. The increasing levels of TDR in migrants could indicate an increase also in their countries of origin, where more efficient surveillance should occur.

## 1. Introduction

In the latest years, the number of migrants has dramatically increased globally [1]. Almost 258 million people live outside their home country [1]. International migrants represent around 3% of the world’s population and have different backgrounds. This phenomenon has contributed to increasing health problems amongst migrants in high-income countries, including vulnerability for HIV acquisition and other sexual health issues [1]. In 2017, migrants account for 41% of newly diagnosed cases of human immunodeficiency virus (HIV) infection in the European Union/European Economic Area (EU/EEA) [2]. Late diagnosis is a feature of the HIV epidemic among migrants, it means that migrants are at risk of low CD4 counts at diagnosis, increased morbidity, mortality, and onward transmission. European surveillance data indicates that some migrant groups are more than twice as likely to be diagnosed late as non-migrants [3].

In Portugal, migrants account for 32% of all new HIV diagnoses [4]. Notably, data suggest that the majority of HIV infections among migrant population occur after migration to Portugal, highlighting an important opportunity for HIV prevention.

The global distribution of subtypes and circulating recombinant forms reflects the complexity of the molecular epidemiology of HIV-1, which might be affected by human mobility, including the different patterns of mobility: local migration, short journey migrants, migration by stages, long-journey migrants and temporary migrants [5]. Migrants are classified as an HIV key population, epidemiologically carrying increased risk, vulnerability and burden of infection due to a combination of socio-economic, biological and structural factors (UNAIDS).

The introduction of highly active antiretroviral therapy (HAART) into developed countries in 1996 has radically changed the clinical outcome of HIV, leading to decreased mortality and morbidity with HIV-1 infection [6,7,8,9]. However, the high replication rate of the virus together with the low fidelity of the viral reverse transcriptase, recombination, and hypermutation altogether are responsible for the presence of high amounts of genetic variation. Whenever viral replication is ongoing in the presence of antiretroviral drugs, these variants that may escape the inhibitory effects of the drugs will be selected inducing the development of antiretroviral drug resistance [10,11,12].

The prevalence of people living with HIV in Portugal has been increasing continuously, and concomitantly the proportion of patients treated with antiretrovirals has also been increasing [4,13].

In the 16th century Portugal colonized several countries, including countries in Africa (Angola, Mozambique, Cabo Verde, Guinea-Bissau), as well as Brazil in South America. HIV molecular epidemiology studies are crucial in these areas to prevent the widespread transmission of drug resistance, especially those lacking systematic surveillance as Portuguese Speaking African Countries (PALOP). The main objective of this study was to analyze the prevalence and patterns of transmitted antiretroviral drug resistance (TDR) and acquired drug resistance (ADR) and the current circulating HIV-1 variants among HIV-1 infected patients migrants originating from Portuguese speaking countries, followed-up in Portuguese hospitals. To achieve this, we combined socio-demographic data, clinical and of viral sequences sampled from migrants followed in Portuguese hospitals and compared it with a dataset of Portuguese sequences available at the same time period of the study.

## 2. Methods

### 2.1. Study Population

The study population included 5177 HIV-1 positive patients independent of their AIDS status. Eligible participants were adults older than 18yo who attended the HIV/AIDS service from Portugal. Patients were included if they had at least one available genotypic resistance profile between January 2001 and December 2017. Overall, 1490 were migrants, and 3687 were Portuguese individuals followed up during the same time aforementioned. Sociodemographic (age, sex, country of origin and attending hospital), clinical (CD4+ T cell count) and virological (viral load) data were linked to partial pol gene sequences (protease, codons 1-99) and reverse transcriptase, codons 1-247). Data was extracted from the central RegaDB resistance database hosted at the Egas Moniz hospital (Lisbon/Portugal) [14] For recorded values of CD4+ cell count and HIV RNA viral load, the most recent measurement of the sample date was defined.

### 2.2. HIV-1 Subtype Assignment

Part of *pol* gene (PR/RT) were performed by different laboratories in whole country using in-house and/or commercial drug resistance tests. HIV-1 subtypes were determined by REGA HIV-1 Subtyping Tool [15,16] software, jpHMM Program (http://jphmm.gobics.de/submissionhiv.html) [17] and Context-based Modeling for Expeditious Typing (COMET, https://comet.lih.lu/) [18].

### 2.3. Drug Resistance Profile

Pol sequences generated by sanger sequencing population were analyzed on Stanford CRP V.6.0 tool to detect for surveillance drug resistance mutations (SDRMs), according to the WHO 2009 SDRM list [8]. The presence of any SDRMs was classified as TDR for epidemiological analysis (https://hivdb.stanford.edu). In order to access Acquired Drug Resistance (ADR), the Genotypic Resistance Interpretation Algorithm of the HIVdb program (http://sierra2.stanford.edu/sierra/servlet/JSierra) was used. The HIVdb program was also used to infer the resistance profile of the HIV-1 sequences and its clinical impact score. The Stanford algorithm comprises mutations contained in the IAS-USA drug resistance mutation list and classifies isolates as susceptible/potential (S), low (L) intermediately (I) or high (H). It was estimated according to the HIVdb Interpretation Algorithm version 8.4 (Stanford University, Palo Alto, CA, USA).

### 2.4. Statistic Analyses

Descriptive statistics for continuous variables of HIV-1 infected individuals subjects were calculated as frequency (percentage) and median Interquartile ranges (IQR:25%-75%). Differences between group were calculated by Mann–Whitney U test (MWT) and the Kruskal-Wallis. Proportions were given with a 95% confidence interval (CI) based on binomial distribution. Differences in proportions were assessed by chi-squared test. we divided patients into 4 groups by date of sampling (2001–2008 vs. 2009–2011 vs. 2012-2014 vs. 2015–2017). Simple logistic regression of global TDR and each class of drugs was performed. Simple and multiple binary logistic regression models were also performed to identify possible factors associated with TDR and ADR. The variables included: age, country, subtype, gender, CD4+, VL and sampling year. Variables with *p* < 0.05 were retained for adjusted analyses. The variables included in the adjusted analysis were age, country of origin, subtype, VL and sampling year. All statistical associations were considered significant if *p* < = 0.05. Statistical analyses were conducted using SPSS and on R.

### 2.5. Ethics Statement

All analyses were performed anonymously. This study was approved by the ethical committee of Egas Moniz hospital (Lisbon/Portugal). All procedures performed in studies involving human participants were in accordance with the ethical standards of the Clinical Research ethical committee of Egas Moniz Hospital (108/CES-2014 – 15-10-2014) and with the Helsinki declaration. It was designed to protect the rights of all subjects involved under the appropriate local regulations.

## 3. Results

### 3.1. Clinical Characteristics of Study Participants

A total of 5177 HIV-1 sequences were included in the analysis and consisted of 1281 (24%) of HIV-1 adult migrants from Portuguese-speaking African countries (PALOP), 209 (4%) from Brazil and 3687 (72%) Portuguese-originated patients, followed up between 2001 and 2017. Overall, 3552 (69%) naïve patients and 1589 (31%) were adhering to a therapeutic regime had complete RT and PR sequences. The number of patients per year varied from 55 to 523 since 2001–2017, including 1839 (35.5%) women and 3294 (63.4%) men with a median age of 39 years (range 32–49). More than 60% of patients had viral load measured and the median plasma HIV RNA was 4.64 log10 copies/mL (3.9–5.2), and the median CD4 count was 281cells/μL (range 128–461 cells/μL). Significant differences between subjects without vs. with previous treatment were found in geographical origin of samples (*p* = 0.044), median of CD4+ T cell counts was 323 (133–498) cells/μL and 244 (121 – 408) cells/μL (*p* < 0.001) and finally HIV viral RNA load was 4.8 (4.1–5.3)log copies/mL and 4.33 (3.6–4.9)log copies/m (*p* < 0.001) on the other hand, there were no significant differences between gender and age. The characteristics of the participants included in the analysis are summarized in Table 1.

We split migrants into two distinct groups: Portuguese speaking African Countries- PALOP and Brazil) and compared it with native population from Portugal. Based on that, migrants from PALOP are older (41yo) than Portuguese (39yo) and Brazilians 34 yo (*p* < 0.001). The epidemic among PALOP is driven by female (53%). However, in the Brazilian migrants and Portuguese groups, females are the minority: 34% and 36%, respectively (*p* < 0.001). The median of CD4+ T cell counts among PALOP was lower 217cells/μL than Brazilian 370 cells/μL and Portuguese 289 cells/μL (*p* < 0.001) (Table 2).

### 3.2. HIV-1 Subtype Determination

A total of 5177 samples were successfully screened for subtype analysis. Viruses were predominantly B (1853, 35.8%), followed by subtype G (1382, 26.7%), Recombinants (663, 12.8%), CRF02_AG (499, 9.6%), subtype C (361, 7%), subtype F1 (185, 3.6%), subtype A (174, 3.4%), subtype D (42, 0.8%) and others (18, 0.4%).

Among migrants from PALOPs, CRF02_AG was the most common genotype (27.1%), followed by URFs (18%), subtype G (17%), subtype C (16.4%), subtype B (8.4%), A (5.5%) and other subtypes (8%). Within migrants from Brazil the most prevalent subtype was B (58.4%), followed by C (12%), recombinants (10%), G (8.6%), F1 (5.7%), and other subtypes (5%). The most prevalent subtype detect among Portuguese patients was B (44%) followed by subtype G (31%), URFs (11.2%), CRF02_AG (4%), subtype C (3.4%) and other subtypes (3.1%). Significant differences between PALOP, Brazil and Portuguese were observed (*p* < 0.001) (Table 2, Figure 1).

### 3.3. Resistance Profile in Drug Naïve (DN) Patients

Among the entire DN population of the study, at least one SDRM was detected in 342 out of the 3552 cases (9.6%, 95% CI: 8.6–10.7) in the HIV-1 polymerase. Mutations associated with NNRTI resistance, present in 186 cases (5.2%, 95% CI: 4.4–5.9), were the most common, followed by mutations associated with NRTI resistance 144 cases (4.1%, 95% CI: 3.4–4.7) and PI resistance 103 cases (2.9%, 95% CI: 2.4–3.5) (Table 2). Within migrants, the overall prevalence of any TDR, NRTI, NNRTI and PI TDR were 9.1%, 4.5% (95% CI: 3.4%-5.9%), 6.4% (95% CI: 5%-8%) and 0.8% (95% CI: 0.4%-1.5%), respectively. In this study, we did not detect significant differences for global TDR, NNRTI and NRTI TDR between migrants and Portuguese, however the prevalence of PI TDR was 3 times higher in Portuguese population compared to Migrants (*p* < 0.001) Table 2. We found higher prevalence of TDR in migrants from Mozambique (17.4%), followed by Angola (10%), Brazil (8.5%), Guinea-Bissau (7.8%), Cape-Verde (7.6%), Sao Tome and Principe (7.1%), but the difference was not significant (*p* = 0.161).

Among migrants the majority (63.4%) of the TDR (*n* = 59) harbored a singleton mutation (NRTI: 16; NNRTI: 35; PI:8). We detected 28% of primary resistance for dual distinct classes of drugs (NRTI/NNRTI) *n* = 26.

Comparisons between patients infected with HIV-1 strains harboring any drug-resistant mutations and those without are shown in Table 3. We observed some differences in the rate of TDR within subtypes. At least one SDRM was observed in 13.7% of all subtype C, 11% of B, 11% of F1, 10.9% of recombinant mosaic sequences and lower than 10% of other subtypes (*p* = 0.005). The logarithmic median of viral load in TDR group were lower (*p* = 0.001) and the median CD4 cell count were slightly higher than those in without TDR group (*p* = 0.024).

The trend of global TDR (OR = 1.078, *p* = 0.007, Figure 2A), transmitted NNRTI resistance (OR = 1.770, *p* = 0.019, Figure 2B) and transmitted NRTI (OR = 2.044, *p* = 0.015, Figure 2C), increased significantly in our study cohort between 2001 and 2017. In contrast, the trend of transmitted PI (OR = 0.835, *p* = 0.571, Figure 2D) resistance remained stable between 2001 and 2017. Among migrants, the risk of TDR increases almost 8% year by year p_trend_ = 0.007.

Overall, among the 144 DN patients with TDR to NRTI (14,2%; 95% CI: 12.1%–16.4%), associated with low to high resistance to a drug according to the Stanford HIV database algorithm v 8.4, the most prevalent mutations (≥10% of individuals harboring mutations) were M184V (*n* = 24; 25%), M41L (*n* = 14; 15%), T215F/Y/S (*n* = 12; 13%). For NNRTIs the most prevalent mutations were K103N (*n* = 35; 37%), Y181C/I (*n* = 15; 17%), G190S/A (*n* = 16; 17%) (K101E *n* = 9; 10%). For PIs, the most prevalent mutation was L90M (*n* = 2; 2.2%) Figure 3.

For NNRTIs, 93 of 1017 sequences (9.14%) had mutations interpreted as moderate and high-level resistance. From those, 6.2% had moderate and high resistance to Efavirenz (EFV) and 2.8% to Rilpivirine (RPV). For NRTI, 82 sequences 8.06% had moderate and High-level resistance, 2.7% resistance to Emtricitabine (FTC), 2.6% to Lamivudine (3TC), 1.6% to abacavir (ABC) and 1.08% to Tenofovir (TDF). For PIs, we did not detect any moderate or High-level resistance (Figure 4).

### 3.4. Resistance Profile in Treated Patients

Among treated patients, the global ADR was detected in 980 out of the 1589 cases (61.6%). Within migrants, the overall prevalence of ADR was 61.5%, similarly to what was found in the autochthonous population (Portuguese patients) (62%) (Table 2). NRTIs resistance mutations were predominantly identified (57.8%), followed by resistance to NNRTIs (45.8%) and PIs (23.6%).

The trend of global ADR (OR = 0.935, *p* = 0.005, Figure 5A), acquired NRTI resistance (OR = 0.916, *p* <0.001, Figure 5C) and acquired PI resistance (OR = 0.826, *p* <0.001, Figure 5D), decreased significantly overtime. On the other hand, the trend of acquired NNRTI (OR = 0.995, *p* = 0.841, Figure 5B) resistance remained stable.

The most prevalent mutations conferring resistance to NRTIs, in treated patients, were M184IV (45.5%) and TAMs, such as T215YF (18%). K103NS (24.9%) and L90M (5.7%) were the most prevalent mutations conferring resistance to NNRTIs and to PIs, respectively (Figure 3).

### 3.5. Drug Resistance Outcome

To explore the predictors of transmitted and acquired drug resistance, we performed multivariable unadjusted and adjusted analysis using age, gender, CD4 count, VL at sample collection, subtype, country of origin, and sampling year. After adjustment, migrants from Mozambique were 1.7 times as likely to present TDR than migrants from PALOP and Brazil. Migrants infected with subtype B had 71% higher chance of harboring TDR. Patients sampled more recently were more likely to present TDR. The same analysis was done for ADR, and after adjustment, age, infection with subtype B, VL (log) and sampling year remained independently associated with ADR (Table 4).

## 4. Discussion

In this study, we analyze the prevalence and trends of TDR and ADR in migrants followed up in Portugal between 2001 and 2017. With a sample size of 1490 migrants and 3687 autochthonous, we could analyze and compare the long-term time trend of TDR/ADR in these populations.

Our population included 1281 (85.9%) migrants from Portuguese speaking African countries and 209 (14.1%) from Brazil. Those countries have long term important relations with Portugal and, except for Brazil, there is very little knowledge in HIV molecular epidemiology and surveillance of drug resistance mutations locally.

Interestingly, the population of PALOPs had a median CD4 count much lower than that of Brazilian migrants and Portuguese population in general, indicating that these immigrants are diagnosed later. Another study conducted in Sweden with African migrants and the Swedish population found that among migrants the CD4 levels were significantly below the general population levels [19]. Another difference observed between PALOPs and Brazilian is the highest proportion of female cases, where once again the epidemic among Brazilian migrants and autochthonous are more related. As already described, migrant populations, largely from Sub-Saharan Africa (SSA), represent a considerable proportion of AIDS cases and HIV infections, especially among female and heterosexual risk factor [20]. However, there were no observed differences between the migrants and the Portuguese population in respect to the viral load median.

A total of 7 subtypes, such as: A, B, C, D, F1, G, H, including CRF02_AG, UFRs, were detected among migrants and the identified subtypes were highly consistent with the ones causing epidemic in their countries of origin [21,22,23]. Consistently, the picture of the HIV epidemic in patients originating from African countries were quite different from autochthonous and from patients originating from Brazil. In patients originating from PALOPs, CRF02_AG (27%) was the most prevalent, followed by URFs (18%), subtype G (17%) and C (16%). On the other hand, among migrants from Brazil, the epidemic is driven by subtype B (58%) followed by subtype C (12%), recombinants (10%) and subtype G (9%). _Consistently, previous studies in Brazil highlight that prevalence of subtype B is around 70–80% through the country followed by subtypes F1, C and recombinants BF1 [24,25,26,27]. Interestingly, we found 9% of Brazilian migrants infected with HIV-1 subtype G. Given the high prevalence of this subtype in Portugal [28] and its rarity in Brazil, this could indicate infections potentially acquired in Portugal, after migration. We are further performing phylogenetic studies to describe transmission networks and get robust information concerning the place of acquisition of infection among migrants.

We observed a TDR rate of 9.1% among the 1017 antiretroviral-naïve PALOP and Brazilian migrants attending Portuguese HIV/AIDS clinics, low to moderate drug-resistance (5%–15%) according to the World Health Organization. This rate is very similar to that found in other sub-Saharan countries, including the neighboring Cameroon with a TDR around 9% [29]. However, the recently published study involving a cohort of migrants from Sub Saharan Africa diagnosed in Sweden showed a slight lower prevalence of TDR (7.1%) than the ones observed in our study [19].

As reported recently by CASCADE collaboration in Europe [30], the prevalence of TDR is trending down, with stable rates in recent years in developed countries [31]. However, herein we find increasing rates of TDR in migrants, especially for TDR mutations conferring resistance to NRTI and NNRTI drug class [32,33,34]. Our multivariate analysis indicated subtype B and country of origin Mozambique as factors associated with higher risk for TDR [35,36].

Among migrants from different Portuguese-speaking countries, migrants from Mozambique showed a high resistance rate of 17.4%, compared to the average of other Portuguese-speaking countries analyzed herein (7–10%). These results are discordant with other studies involving patients from Mozambique between the years 2007 and 2009 that showed a low prevalence of 5% of primary drug resistance based on WHO mutation list [37,38]. These differences may be related to four hypotheses: 1) The rate of TDR involving increased, since the studies conducted in this area are older. 2) The coverage of adults and children receiving ART in the recent years was much higher than in the last years (UNAIDS). 3) Poor adherence to ART in Mozambique is a problematic issue [39]. 4) These migrants became infected post-migration and might have been infected by a virus with resistance mutations within the community Mozambique migrants in Portugal.

While examining the presence of resistance by drug class, we found that there was a significant difference in the prevalence of drug-resistant mutations between the three antiretroviral classes (NRTI, 4.5%; NNRTI, 6.4%; and PI, 0.8%) These SDRMs had an impact on the efficacy of several treatment strategies, especially efavirenz (EFV) and lamivudine (3TC), respectively. Between 2010 and 2017, EFV and 3TC were recommended for first line treatment in Portugal. This was also the most frequent NRTI and NNRTI-resistance mutation (M184V and K103N) among pre-treated patients and was previously reported among antiretroviral-naïve patients living in Portuguese Speaking countries [38,40,41]. A low prevalence of TDR to PIs within migrants reflects the scarce administration of this antiretroviral as first line in their countries of origin. However, the prevalence of PIs SDRMs in autochthonous patients from Portugal is almost 4 times higher than in migrants, due to the fact that they were recommended for first line in Portugal and Europe. As such, this higher rate can be a reflection of a long term selective pressure of PI drugs in the population of autochthonous patients, that selected for resistant mutations that continue to be transmitted.

In treated migrant patients followed up in Portugal, we found that the rate of acquired drug resistance was 61.5%. This high rate was coincident with the rate of ADR among autochthonous Portuguese patients (62%). Other European studies have revealed a high rate of ADR around 64% [42,43]. This ADR rate in migrants probably reflects long term follow up in portuguese hospitals and, therefore, is similar to autochtonous patients followed up in the same hospital. It also suggests that adherence levels should be roughly similar in both populations.

Infection with subtype B was shown to be significantly associated with a higher risk of ADR in the univariate and multivariate analysis, as well as age and viral loads.

Our study has a number of weaknesses and limitations. Missing information about risk group limits interpretations about the importance of risk groups for transmission and acquisition of drug resistance. For example, the higher rate of DR found in subtype B infected patients could be associated with the higher proportion of men who have sex with men infected with this subtype [44]. Another limitation of our study was the fact that migrants some countries were underrepresented compared to others: Sao Tome and Principe, Mozambique, and Brazil. The number of patients originating from Mozambique was also quite low (*n* = 121). As such, and although TDR in migrants from Mozambique was found to be higher than that observed in the other migrants it is important to conduct other studies with a larger number of patients to better understand this issue.

This work highlights the importance of characterizing the ARV drug resistance profile in migrants. Through this report, it will be possible to better understand how HIV drug resistance is circulating in migrant populations and thus develop new strategies to better manage the epidemic.

## Figures and Tables

**Figure 1 viruses-12-00268-f001:**
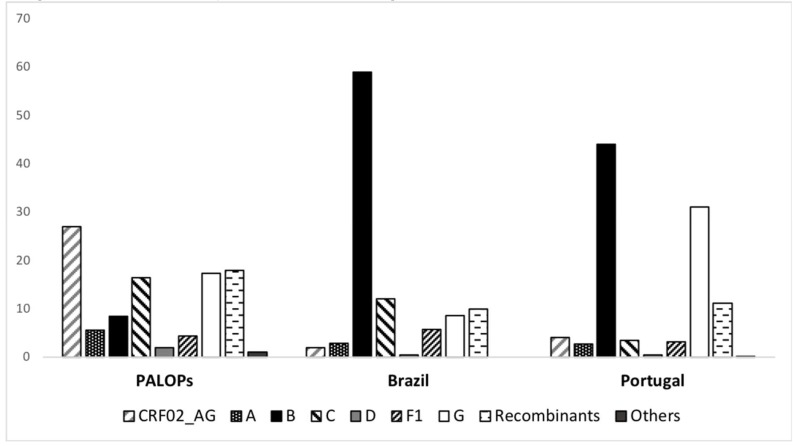
Subtype distribution stratified by country of origin of the patient. Portuguese Speaking African Countries (PALOPs) *n* = 1281, Brazil *n* = 209 and Portugal *n* = 3687.

**Figure 2 viruses-12-00268-f002:**
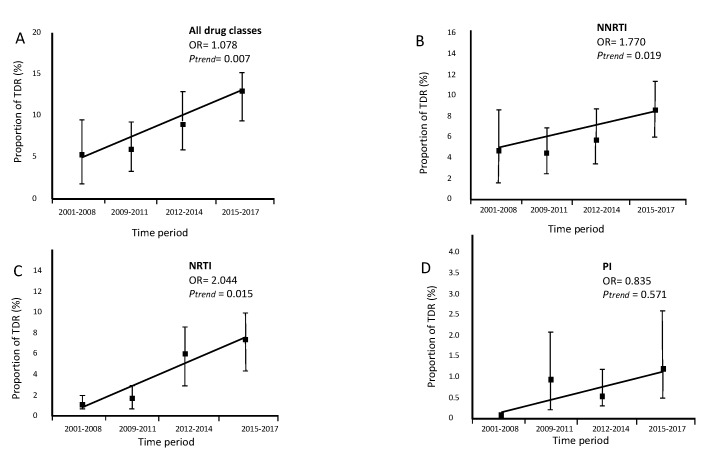
Trends of proportion of transmitted drug resistance (TDR) in migrants followed up in Portuguese hospitals (2001–2017). N(2001–2008) = 137, N(2009–2011) = 325, N(2012–2014) = 228, N(2015–2017) = 327. (**A**) Trend for overall TDR. (**B**) Trend for transmitted nucleoside reverse transcriptase inhibitor (NRTI) resistance. (**C**) Trend for transmitted NNRTI resistance. (**D**) Trend for transmitted protease inhibitors (PI) resistance.

**Figure 3 viruses-12-00268-f003:**
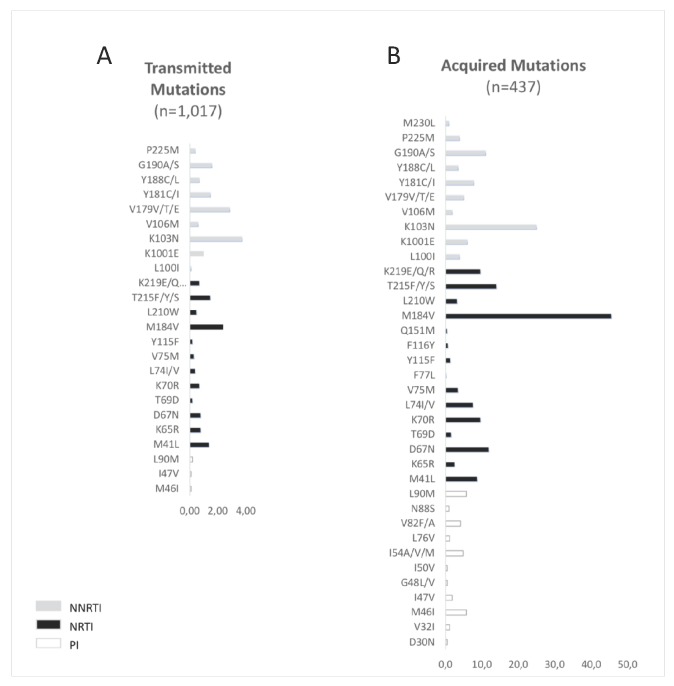
Proportion of human immunodeficiency virus (HIV) mutations in (**A**) drug-naïve patients and (**B**) treated patients between 2001 and 2017.

**Figure 4 viruses-12-00268-f004:**
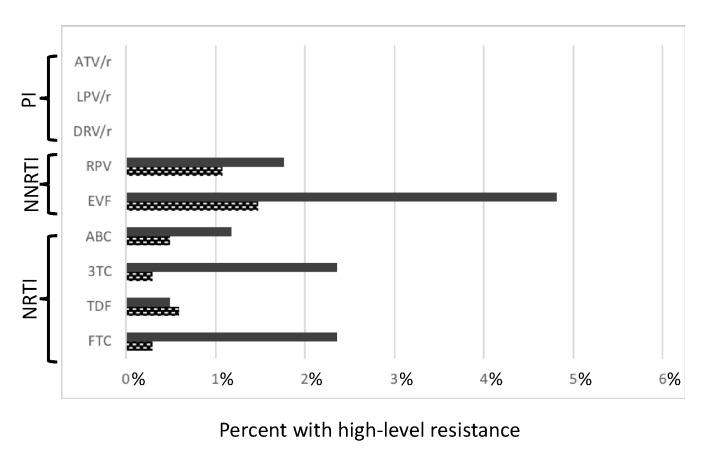
Predicted phenotypic resistance (Standford scores) for antiretroviral drugs currently recommended as first line therapy in Portugal for drug naïve patients. Hight level resistance (Stanford scores > 3; bold bars indicate a score of 5. Checked bars indicate a score of 4, related with a first line antiretroviral drugs recommended by the European AIDS clinical society. Abbreviations: NRTIs = Nucleoside reverse transcriptase inhibitors; NNRTIs = Non-nucleoside reverse transcriptase inhibitors; PIs = Protease inhibitors; FCT = Emtricitabine; TDF = Tenofovir; 3TC = Lamivudine; ABC = Abacavir; EVF = Efavirenz; RPV = Rilpivirine; DRV/r = Darunavir; LPV/r = Lopinavir; ATV/r = Atazanavir.

**Figure 5 viruses-12-00268-f005:**
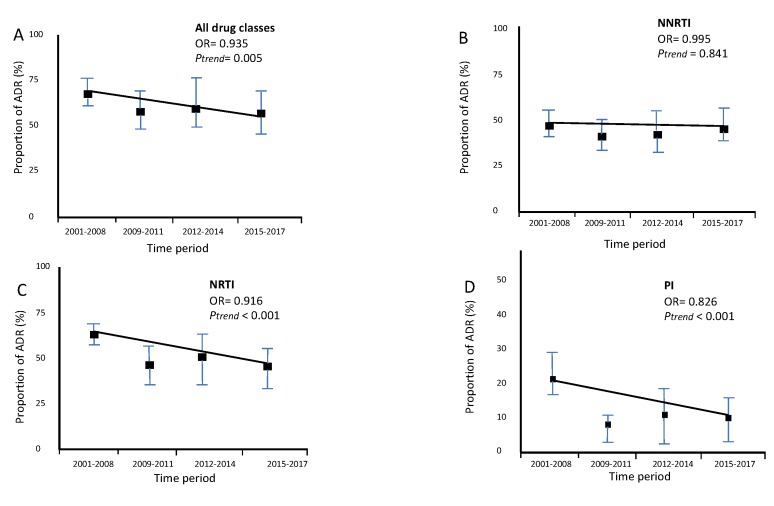
Trends of proportion of ADR in migrants followed up in Portuguese hospitals (2001–2017). N(2001–2008) = 173, N(2009–2011) = 127, N(2012–2014) = 51, N(2015–2017) = 86. (**A**) Trend for total ADR. (**B**) Trend for Acquired NRTI resistance. (**C**) Trend for Acquired NNRTI resistance. (**D**) Trend for Acquired PI resistance.

**Table 1 viruses-12-00268-t001:** Characteristics of migrants and Portuguese population followed up in Portugal, 2001–2017.

		Treatment Naive			Treatment Experienced			*p*
**Characteristics**	**Overall** **(*n* = 5177)**	**Total** **(*n* = 3552)**	**No TDR** **(*n* = 3210)**	**With TDR** **(*n* =342)**	**Total** **(*n* = 1589)**	**No ADR** **(*n* = 609)**	**With ADR** **(*n* = 980)**	
**Age**								
**median ± IQR**	39 (32–489)	39 (31–49)	39 (31–48)	40 (31–49)	40 (33–47)	39 (33–46)	40 (34–48)	0.088
**Gender**								
**Number (%)**								0.077
**Male**	3294 (63.4)	2288 (64.4)	2060 (64.2)	228 (66.7)	983 (61.9)	352 (57.8)	631 (64.4)	
**Female**	1839 (35.5)	1233 (34.7)	1124(35)	109 (31.9)	593 (37.63)	248 (40.7)	345 (35.2)	
**Unknown**	44 (0.8)	31 (0.9)	26 (0.8)	5 (1.5)	13 (0.8)	9 (1.5)	4 (0.4)	
**geographical origin**								**0.004**
**PALOPS**	1281 (24.7)	853 (24)	774 (24.1)	79 (23.1)	395 (24.9)	153 (25.1)	242 (24.7)	
**Brazil**	209 (4)	164 (4.6)	150 (4.7)	14 (4.1)	42 (2.6)	15 (2.5)	27 (2.8)	
**Portugal**	3687 (71.2)	2535 (71.4)	2286 (71.2)	249 (72.8)	1152 (72.5)	441 (72.4)	711 (72.6)	
**HIV-1 RNA level (Log10 copies/mL)**								
**median ± IQR**	4.64 (3.9–5.2)	4.8 (4.1–5.3)	4.8 (4.2–5.3)	4.64 (4.09–5.1)	4.33 (3.6–4.9)	4.65 (3.8–5.2)	4.15 (3.5–4.7)	***p*** **< 0.0001**
**CD4 countss (cells/** **µ** **L)**								
**median ± IQR**	281 (128–461)	323(133–498)	319 (129–493)	3170 (209–549)	244 (121–408)	229 (125–386)	248 (118–420.7)	***p* < 0.0001**

**Table 2 viruses-12-00268-t002:** Characteristics of migrants versus Portuguese population followed up in Portugal, 2001–2017.

	Overall (*n* = 5177)	Host Country (*n* = 3687)	Migrants (*n* = 1490)		*p*
**Variables**		**Portugal**	**Brazil**	**PALOP**	
**Age_median ± IQR_**	39 (32–48)	39 (32–47)	34 (28–41)	41 (33–50)	**<0.0001**
**Gender_(number (%)_**					**<0.0001**
Male	3294 (64)	2563/3687(70)	137/209 (66)	594/1267 (46.9)	
Female	1839 (36)	1095/3687 (30)	71/209 (34)	673/1267 (53.1)	
**Subtype_(number (%)_**					**<0.0001**
02_AG	499/5177 (9.6)	148/3687 (4)	4/209 (1.9)	347/1281 (27)	
A	174/5177 (3.4)	98/3687 (2.7)	6/209 (2.9)	70/1281 (5.5)	
B	1853/5177 (35.8)	1623/3687 (44)	122/209 (59)	108/1281 (8.4)	
C	361/5177 (7)	126/3687 (3.4)	25/209 (12)	210/1281 (16.4)	
D	42/5177 (0.8)	15/3687 (0.4)	1/209 (0.5)	26/1281 (2)	
F1	185/5177 (3.6)	116/3687 (3.1)	12/209 (5.7)	57/1281 (4.4)	
G	1382/5177 (26.7)	1143/3687 (31)	18/209 (8.6)	221/1281 (17.3)	
Others	18/5177 (0.4)	5/3687 (0.1)	0	13/1281 (1)	
Recombinants	663/5177 (12.8)	413/3687 (11.2)	21/209 (10)	229/1281 (17.9)	
**ANY_TDR_(number (%)_**	342/3552 (9.6, 8.6–10.7)	249/2535 (9.8, 8.6–10.9)	14/164 (8.5, 4.2–10.7)	79/853 (9.2, 7.2–11.1)	0.795
**NRT_(number (%)_**	144/3552 (4.1, 3.4–4.7)	98/2535 (3.9, 3.1–4.6)	4/164 (2.4, 0.9–6.1)	42/853(4.9, 3.4–6.3)	0.224
**NNRT_(number (%)_**	186/3552 (5.2, 4.4–5.9)	121/2535 (4.8, 4–5.6)	90.055 (5.5, 2.9–10)	56/853 (6.5, 5–8.4)	0.125
**PI_(number (%)_**	103/3552 (2.9, 2.4–3.5)	95/2535 (3.7, 2.9–4.4)	2/164 (1.2, 0.4–2.9)	6/853 (0.7, 0.1- 1.2)	**<0.0001**
**ANY _ADR_(number (%)_**	980/1589(61.7, 58.6–63.4)	711/1152(62, 59.2–64.8)	27/42 (64, 49.5–78.5)	242/395(61, 56.2–65.8)	0.676
**VL_median ± IQR_**	4.6 (3.9–5.2)	4.66(3.9 – 5.2)	4.7 (4.04– 5.28)	4.5 (3.8–5.2)	0,29
**CD4_median ± IQR_**	281(128–461)	289 (130 – 470)	370(130–529)	217 (113–369)	**<0.0001**

Quantitative variables values are expressed as median and IQR (25–75 percentiles) and quantitative variables are displayed as number of cases (percentage and Confidence Interval-95% in parentheses). HIV subtypes were determined based only on a partial pol sequence. TDR (Transmitted drug resistance), ADR (Acquired Drug Resistance), VL (Viral load in log10), CD4 (cd4 T cell count).

**Table 3 viruses-12-00268-t003:** Characteristics of naïve-treated patients versus treated patients.

Variables	Naïve			Treated		
	Any TDR	Susceptible	*p*	Any ADR	Susceptible	*p*
**Age_median ± IQR_**	41.2 (31.7–49)	40.4 (31–48)	0.47	41.2 (34–48)	40 (33–46)	**0.0225**
**Gender_(number (%)_**			0.281			**0.018**
Male	228/2288 (9.9, 8.6.11.1)	91.1%		631/983 (64.1, 61.1–67.1)	39%	
Female	109/1233 (8.8, 7.2–10.3)	90.1%		345/593 (58.1, 54.1–62)	37%	
**Countries_(number (%)_**			0.1614			0.416
Angola	21/212 (9.9, 5.88–13.9)	90.1%		92/144 (63.8, 55.9–71.6)	36.2%	
Brazil	14/164 (8.5, 4.23–12.7))	91.5%		27/42 (64, 49.4–78.5)	36%	
Cape Verde	14/186 (7.6, 3.79–11.4))	92.4%		72/107 (67.2, 58.3–76.1)	32.8%	
Guinea-Bissau	24/307 (7.8, 4.8–10.8)	92.2%		49/95 (51.5, 41.4–61.5)	48.5%	
Mozambique	16/92 (17.4, 9.6- 25.1)	82.6%		17/29 (58.6, 40.6–76.5)	41.4%	
Sao Tome and Principe	4/56 (7.1, 3.7–13.8)	92.9%		12/20 (60, 38.5–81.4)	40%	
Portugal	249/2535 (9.8, 8.6–10.9)	90.2%		711/1152 (61.7, 58.9–64.5)	38.3%	
**Subtype_(number (%)_**			**0.005**			**0.02**
02_AG	30/365 (8.2, 4.2–10.7)	92.5%		61/120 (50.2, 41.2–59.1)	51.2%	
A	7/144(4.9, 1.3–8.2)	95.2%		16/28 (57, 38.6–75.3)	43%	
B	141/1278 (11, 9.8–13.2)	88.5%		385/570 (67.2, 63.3–71)	32.8%	
C	36/261 (13.7, 9.6–18)	86.2%		54/92 (58.6, 48.5–68.6)	41.4%	
D	3/31(9.6, 0.7–19.9)	90.4%		8/11 (72.7, 46.3–99)	27.3%	
F1	17/153 (11, 6–16)	89%		18/31 (58, 40.6–75.3)	42%	
G	57/843 (6.8, 5.1.8.5)	93.2%		317/537 (59, 54.8–63.1)	41%	
Recombinants	51/464 (10.9, 8.1–13.2)	89.3%		119/195 (61, 54.1–67.8)	39.8%	
**VL_median ± IQR_**	4.6 (4.0 – 5.1)	4.7 (4.2– 5.4)	**0.011**	4.1 (3.5 – 4.7)	4.5 (3.8–5.2)	**<0.0001**
**CD4_median ± IQR_**	388 (209 – 549)	345 (129–493)	**0.024**	291 (118 – 420)	287 (125–386)	0.452

**Table 4 viruses-12-00268-t004:** Influence of baseline factors and treatment modification to developed Transmitted Drug Resistance (TDR) and Acquired Drug Resistance (ADR).

	Naive		adjusted		Treated		Adjusted	
Variables	Sig	OR (95% C.I Expected)	Sig	OR (95% C.I Expected)	Sig	OR (95% C.I Expected)	Sig	OR (95% C.I Expected)
**Gender (Female^(a)^/Male, n = 1475)**	0.913	1.024(0.666–1.576)			0.148	1.332(0.903–1.966)		
**Age (*n* = 1465)**	**0.026**	1.021(1.0029–1.002)	0.938	1.001(0.985–1.016)	**0.001**	1.032(1.013–1.053)	**0.003**	1.043(1.014–1.072)
**Country (*n* = 1490)**								
**Angola (No^(a)^/Yes)**	0.666	1.119(0.671–1.867)			0.482	1.159(0.767–1.752)		
**Brazil (No^(a)^/Yes)**	0.768	0.914(0.505–1.657)			0.724	1.138(0.587–2.208)		
**Cape Verde (No^(a)^/Yes)**	0,398	0.775(0.429–1.401)			0.162	1.389(0.877–2.200)		
**Guinea (No^(a)^/Yes)**	0.335	0.788(0.485–1.280)			**0.025**	0.591(0.373–0.935)	0.932	1.030(0.521–2.036)
**Mozambique (No^(a)^/Yes)**	**0.005**	2.319(1.289–4.172)	**<0.001**	2.71(1.605–2.730)	0.737	0.877(0.408–1.886)		
**Sao Tome (No^(a)^/Yes)**	0.594	0.754(0.266–2.133)			0.884	0.934(0.374–2.334)		
**Subtype (*n* = 1490)**								
**CRF02_AG (No^(a)^/Yes)**	0.320	0.766(0.453–1.290)			**0.007**	0.521(0.324–0.839)		
**A (No^(a)^/Yes)**	0.180	0.376(0.090–1.570)			0.670	0.825(0.340–2.002)		
**B (No^(a)^/Yes)**	**0.027**	1.780(1.067–2.969)	**0.022**	1.719(1.082–2.730)	**0.011**	2.169(1.191–3.950)	**0.019**	2.687(1.175–6.143)
**C (No^(a)^/Yes)**	0.141	1.484(0.877–2.510)			0.437	0.8707(0.470–1.386)		
**D (No^(a)^/Yes)**	0.599	0.580(0.076–4.407)			0.324	2.2183(0.455–10.803)		
**F (No^(a)^/Yes)**	0.061	2.045(0.967–4.323)			0.254	0.525(0.173–1.589)		
**G (No^(a)^/Yes)**	0.112	0.545(0.258–1.151)			0.168	1.407(0.866–2.286)		
**Rec (No^(a)^/Yes)**	0.377	0.752(0.400–1.414)			0.701	1.102(0.670–1-813)		
**CD4 (*n* = 375)**	0.355	1.001(0.999–1.003)			0.231	1.000(0.999–1.003)		
**VL(731)**	0.294	0.803(0.532–1.210)			**0.002**	0.634(0.477–0.842)	**0.001**	0,600(0.446–0.808)
**Sampling year**	**0.003**	1.085(1.029–1.145)	**0.007**	1.078(1.021–1.139)	**0.005**	0.935(0.892–0.979)	**0.001**	0.821(0.734–0.919)

Logistic regression of variables at the genotyping collection associates to transmitted and acquired drug resistance. The *p* values highlighted in bold in the simple analysis were retained for the adjusted analysis

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
