# Peer review of "Molecular Epidemiology of HIV-1 Infected Migrants Followed Up in Portugal: Trends between 2001–2017"

_viruses, 2020, doi:10.3390/v12030268_

Round 1
Reviewer 1 Report
This is a well-presented paper that focuses on the distribution of HIV subtype and prevalence of drug resistance for different ARV classes in Portugal. The study is sound, clearly written, and well referenced. It is clearly of interest to others in the field, particularly those interested in how patterns of migration contribute to the interplay between the very different HIV epidemics in Europe v Africa.
I have one specific concern: the presentation of country of origin = Mozambique as a significant association with drug resistance is not strongly justified, due to potentially tiny sample size (as mentioned in the Discussion, line 323). No sample size is shown for this subgroup in the tables, which makes me suspicious of the P values. There were approx 3,000 Mozambique migrants living in Portugal in 2015 according to readily accessible figures, eg https://doi.org/10.1016/j.fsigss.2017.09.134 This number may be somewhat larger in 2017+ but nonetheless, is likely small - what share of this population are HIV+? I have concerns that singling out migrants from Mozambique as a risk group for drug-resistant HIV in Portugal may not be ethical. At the very least, the implication of this statement need to be thought through in detail.
Minor comments:
- Subtype distributions are poorly visualised, particularly as the discussion states that the subtypes are more similar to the countries of origin than to what is circulating in Portugal. Please show these results visually, eg. with stacked bar plots showing the subtype mix in each population, in Portugal in general, and ideally in the countries of origin (as far as possible). The presentation of this data in the table is extremely difficult to take in.
- How is Ptrend calculated in figure 1? I could not find an indication of this in the methods. Also please correct the methods to indicate which tests were used where, eg were *both* Fisher's exact test and chi-squared really used? If not, which test does each P value refer to? Please state this in legends as well, for all tables and figures.
- line 193 - "of DN mutations with mutations in..." - I don't understand what this means, is there a typo?
- line 206 - "did not detected" should be "did not detect" (some other typographical errors throughout need correction as well)
- Figure 4 (c) and (d) - the trend is clearly dominated by an unusually high first data point, with all subsequent data points remaining flat. Please comment on this and justify your statistic (again, it's not clear what Ptrend means)
- Table 4: please include the number of individuals in each class. Also, which variables were retained for the adjusted analysis? Methods say those with p<=0.5, but please indicate exactly what was done.
Reviewer 2 Report
The manuscript by Pimentel et al. describes the analyses of HIV-1 transmitted drug resistance mutations (TDRs) in 5177 immigrants from Portuguese speaking countries in Portugal. The study is based on a significant dataset, the analyses have been carried out satisfactorily and the conclusions are backed by the results. Therefore, I only have some minor comments. However, the manuscript must be checked for English grammar, because quite a few language mistakes are made. Also, caps pop up or are lost throughout (see e.g. line 206, line 313…..). Figures and tables are clear.
Comments:
Line 59: maybe ‘enormous’ is somewhat exaggerative
Line 60: likely ’that’ should be inserted after ‘’these variants’
Section 2.5: the location of the Egas Moniz hospital (city/country) should be added
Line 136: ‘is driven by females’
Line 169: ‘albeit there was no statistical significance’ is not correct English
Line 176: where naturally occurring polymorphisms, which may explain the relatively high number of SDRM in subtype C, taken into account by the analysis methods?
Line 206: ‘we did not detect any moderate…..’
Line 266: with ‘autochthonous’ the Portuguese population is meant?
Line 293-294: is it known whether or not cART prescription or availability has changed in Mozambique in recent years to explain the rising TDR prevalence?
The Discussion section is not always an easy read, e.g. line 306 should certainly be rewritten (‘due to the fact that the main PIs were recommended for first line-, Lopinavir’ etc).
